# Investigation on Plastic Flow Behaviors of FCC Polycrystalline Aluminum under Pre-Cyclic Tension-Compression Loading: Experiments and Crystal Plasticity Modeling

**DOI:** 10.3390/nano11092397

**Published:** 2021-09-14

**Authors:** Damin Lu, Keshi Zhang, Guijuan Hu

**Affiliations:** 1Key Laboratory of Disaster Prevention and Structural Safety/Guangxi Key Lab Disaster Prevention and Engineering Safety, College of Civil and Architectural Engineering, Guangxi University, Nanning 530004, China; ludamin@st.gxu.edu.cn (D.L.); zhangks@gxu.edu.cn (K.Z.); 2School of Landscape Architecture, Zhejiang A & F University, Hangzhou 311300, China

**Keywords:** yield surface, plastic flow, crystal plasticity, polycrystalline aluminum

## Abstract

The plastic flow behaviors of FCC polycrystalline aluminum after pre-cyclic tension-compression deformation are mainly investigated in tension–torsion stress space by the physically based crystal plasticity model introducing a back-stress. A global finite element model (GFEM) constructed of sufficient grains was established to simulate the same-size thin-walled tube specimen constrained and loaded as the experiments of yield surfaces. The computational results showed that the shape of subsequent yield surfaces and the plastic flow directions directly depended on the given offset strain levels and the applied re-loading paths under different pre-cyclic deformations. The angle deviation between the plastic flow direction and the theoretical orthogonal direction further indicated that there was a large difference between them in the inverse pre-straining direction, but the difference was negligible in the pre-straining direction. From the influence of the anisotropic evolution of the subsequent yield surfaces on plastic flow, we found that the plastic normality rule followed the smooth yield locus; conversely, the significant non-associated flow was attributed to the distorted yield locus. Furthermore, it was also demonstrated that the anisotropic evolution and the plastic flow trend of the subsequent yield surfaces obtained by experiments can be better reproduced by the crystal plasticity model.

## 1. Introduction

During a sheet forming operation, sheet metals generally exhibit significant plastic anisotropy due to rolling and annealing operations [1,2]; moreover, the plastic anisotropy causes directional dependency of the yield stress, and its degree of anisotropy can be geometrically characterized by the distortion and the non-associated flow rule of the yield surfaces. The plastic anisotropy of materials has tremendous influence on material flow, wrinkling behavior, forming limit, springback, and failure properties of the sheet metal [3,4]; further, such material properties require accurate estimation, which depends on a precise knowledge of the yield surface and plastic flow rule [4]. In general, the material models are dominated by three relations: a yield criterion characterizing a yield surface that defines the critical states between the elastic and elastoplastic ranges, flow rule presenting the direction of the plastic strain rate, and a strain-hardening rule describing the yield surface evolution [5,6]. For the repeated loading and unloading case, the direction change of plastic strain accumulation results in the “sharp corner” direction shift of yield surfaces in the loading direction [7]. Therefore, it is imperatively necessary to develop constitutive models capable of describing the yield surface and the direction of the plastic strain rate under complex cyclic loading history. Namely, more accurate yield criterion and plastic flow theories play a crucial role in exploring the influence of physically based anisotropy on the yield surface [2,8].

Many experimental investigations were conducted to indicate the geometrical shapes of yield surface distortion with the remarkable corner effects in the pre-strain direction [9,10] and non-normality of plastic flow on the smooth yield surface [11,12]. At the same time, two main material constitutive models were employed to predict the anisotropic yielding and plastic flow behaviors of polycrystalline materials: the phenomenological approach introducing the anisotropic yield functions with associated or non-associated flow rule [13,14,15] and the micromechanical approach based on the polycrystalline plasticity models [16,17,18].

For the phenomenological approach, various anisotropic yield functions based on the associated flow rule (AFR) or non-associated flow rule (non-AFR) to account for the anisotropy of yield surface evolution, Bauschinger effect, and strain-hardening behavior have been proposed [2,16,19,20]. The earliest and most widely used yield criteria under the associated flow rule with a single function in a quadratic or a non-quadratic form to describe both yield behavior and the direction of plastic strain were initiated by Hill [21] and Hosford [22], and the yield locus based on these yield criteria was only restricted to the plane stress states excluding shear stress terms and orthotropic symmetry assuming principal stress axes superimposed onto orthotropic axes [3,4,19]. Hill’s yield criterion was further extended by Lin and Ding [23]. To overcome the limitation stated above, the tricomponent and a six-component yield criterion by coupling shear and normal stress components was established to describe the yield surfaces under a full plane stress condition by Barlat et al. [19]. The evolution of the yield surface depends significantly on the deformation history; however, the endochronic theory is inherently suited to describe the responses in general loading histories [24]. The original endochronic theory proposed by Valanis [24,25,26,27] did not use the concept of a yield surface, which deals with the elastic–plastic response of materials in terms of the memory kernel functions that lead to an implicit derived result of a yield surface. The deformation history is described with respect to intrinsic time defined as the path length in the strain space and the rate equations through hereditary function [28]. Further, an endochronic plasticity theory was developed by Yeh et al. [29] to describe the distortion of yield surface by the combination of two ellipses with different aspect ratios governed by the aspect ratio function. Moreover, the non-AFR models with the non-normality of plastic flow as experimental evidence to describe the anisotropic plastic strain rates and yield behavior with two different functions were proposed by Stoughton and Yoon [30]. According to the classical theories of plasticity, the direction of plastic strain is normality or non-normality due to the yield surface evolution exhibiting an ellipse or a rough blunt-nosed corner under proportional loading paths [8], namely, the yield functions and plastic flow rules are either inadequate or inaccurate for describing the non-normality of plastic flow with the complicated sharper curvature of the yield surface obtained with the polycrystalline plasticity theory as experimental observations. However, the yield criterion and flow rule lack a physical basis and the microscopic mechanism to account for the microstructural effects, leading to the difficulties and limitations in the application of the phenomenological models in complex stress conditions.

For the micromechanical approach, since the essential physical mechanisms of anisotropic properties are mainly caused by the crystallography of slip, the texture, or the preferred grain orientation, the yield surfaces are calculated with polycrystalline plasticity models on the basis of the plastic deformation mechanisms using some averaging schemes [18]. Barlat et al. [4] analyzed the effect of crystallographic texture on planar anisotropic yield surfaces using the Taylor/Bishop and Hill model based on the pencil glide system of polycrystalline plasticity. Schurig et al. [31] used the classical Taylor–Lin polycrystal model and a viscous grain model to simulate the development of a yield vertex for different strain processes. Beradai et al. [8] used the self-consistent approach to effectively simulate a very complex evolution of subsequent yield surfaces for real FCC metals under complex loading path. The Taylor model is based on the assumption of the same macroscopic strain state for individual grain in the aggregate. Indeed, in addition to the simplicity, Taylor-based and self-consistent models cannot account for the effect of the grain shape and their interactions. For the solving of these simplified problems, the crystal plasticity finite element method (CPFEM) is well suited to consider the microstructure effects and their interactions of each grain under the displacement continuity assumption of the finite elements at the grain boundaries [32]. Canova et al. [33] pointed out that crystallographic texture gives rise to the anisotropic behavior of yield surfaces for textured materials by polycrystal plasticity simulation. Cazacu et al. [34,35] introduced the texture and grain shape in the modelling of the yield surface using crystal plasticity associated with the representative volume element (RVE). Primary engineering objectives of a crystal plasticity finite element method (CPFEM) applications in macroscopic forming simulations are the prediction of the precise material shape after forming optimization of material flow, elastic spring-back, forming limits, and texture evolution of the formed part [36,37,38,39]. Therefore, it was demonstrated that micromechanical approach based on the crystallography of slip, the texture, or the preferred grain orientation can be used to predict the complicated anisotropic evolution of the yield surfaces as experimental results more accurately than the phenomenological approach under complex loading paths [17,18].

In this study, new results concerning the yield surfaces and the plastic flow direction under cyclic loading paths are presented and are investigated by the crystal plasticity model. A crystal plasticity finite element (CPFE) scheme is established with a new GFEM method, illustrated in Section 4.1 in detail. The dependence of the yield surfaces and the plastic flow directions on unloading point position, pre-strain directions, and the levels of yield definitions are investigated in detail by computational simulation. Furthermore, the capabilities of the models are verified by comparing the plastic flow direction of the yield surfaces simulated by the crystal plasticity model with experimentally observed results using the statistical methods.

## 2. Material and Experimental Procedure

### 2.1. Experimrntal Material and Specimen

The experimental material was commercially pure aluminum with the purity of 99.89% weight. Its chemical composition and basic mechanical properties are given in Table 1. The cross-sectional EBSD map, the grain size distribution, and pole figure observed by electron backscatter diffraction (EBSD) technique are presented in Figure 1. The initial microstructure of the material contains a large number of grains, and the average grain size was about 65 μm, as shown in Figure 1a,b. It can be further seen that the pure aluminum had a weak texture and the grain orientations were almost random from the EBSD orientation map as per the grain orientation represented by the different colors in Figure 1a and pole figure showing its maximum intensity in 3.93 times random in Figure 1c. The tests were carried out by the MTS809 electro-hydraulic servo testing machine (MTS Systems Corporation, Eden Prairie, MN, USA), applying the tension and torsion load at the two gripping ends of the specimen, and an extensometer with a 25 mm gauge length was mounted on the outer surface of the gauge section of specimens to measure the axial and torsion strain. The geometry size of thin-walled tube specimens of aluminum materials were machined for the experiments shown in Figure 2, and the $n_{1}$, $n_{2}$, and $n_{3}$ axes were set along the axial, thickness, and circumferential directions, respectively.

### 2.2. Experiment Produce

The entire loading process of determining subsequent yield is described according to three steps in detail in this section. First, the specimen underwent the pre-cyclic deformation of 30 cycles to achieve the saturated stress–strain state under symmetrical tension-compression loading at the strain amplitude of 0.3% and remained the constant strain for 2 min corresponding to the respective unloading starting points (A (0.3%), B (0.12%), C (0%), or D (−0.3%)) on the hysteresis loop to eliminate the effect of strain rate or viscoplasticity caused by pre-loading [8]. Then, the specimen was subsequently unloaded to the unloading endpoint (O_A1_, O_B1_, O_C1_, or O_D1_) within the elastic domain or the reverse yield endpoint (O_A2_) from unloading the starting point on the saturated stress–strain hysteresis curve by the stress-controlled mode, namely, the five different unloading paths were A (0.3%) → O_A1_, A (0.3%) → O_A2_, B (0.12%) → O_B_, C (0%) → O_C_ or D (−0.3%) → O_D_, as shown in Figure 3a. Finally, the specimen was reloaded to probe the yield point corresponding to different target offset strain along a prescribed tension–torsion proportional reloading path, as shown in Figure 3b.

Here, the target offset strain $\Delta E_{\mathrm{offset}}^{p}$ is prescribed as 20 μ, 50 μ, 100 μ, 200 μ, 600 μ, and 1000 μ under each reloading path as shown in Figure 3b. In this study, the multiple-specimen strategy was chosen, for which the yield pointed along a specified tension–torsion proportional loading path by different offset strain definition, determined using a thin-walled tubular specimen; therefore, a complete yield surface could be obtained using several same specimens under the reloading paths at intervals of 15° ranging from 0° to 180°, as shown in Figure 3b.

In this study, the unloading stiffness method was adopted to determine the yield point; since the unloading stiffness **E**_eff_ decreased as plastic deformation proceeded during the reloading [9,10], the determination of yield points were affected by the unloading stiffness. To capture yield points more accurately at different offset strains along a reloading path, we fitted the unloading stiffness **E**_eff_ by the linear part of the equivalent stress–strain curves during the unloading process of gradual reloading and unloading, as shown in Figure 4a. During gradual reloading and unloading, as soon as an accumulated effective plastic strain increment $\Delta E_{\mathrm{offset}}^{p}$ approached the prescribed target offset strain value ($\Delta E_{\mathrm{offset}}^{p}$ = 20 μ, 50 μ, 100 μ, 200 μ, 600 μ, or 1000 μ, $\mu =10^{-6}$), the intersection point of the straight line $Y=E_{\mathrm{eff}}(X-\Delta E_{\mathrm{offset}}^{p})$ parallel to the unloading stiffness **E**_eff_ and the equivalent stress–strain curve was defined as the subsequent yield point represented by **Y** in Figure 4b. By repeating this method, we were able to determine the subsequent yield points corresponding to each prescribed target strain along a reloading path.

### 2.3. Determining the Orthogonal Direction and Plastic Strain Incremental Direction of Yield Point

For the investigated plastic flow behavior for polycrystalline aluminum, we assumed that the stress space was consistent with the plastic strain space. Figure 2 shows the geometric dimensions of the test specimen. The nominal tension stress σ and shear stress τ on the cross-section of the gauge length range under the combined tension–torsion loading were set along the $n_{1}$ axis and $n_{3}$ axis, respectively, and the direction of the nominal strain was assumed to be coincided with that of the stress. The schematic illustration of plastic strain incremental direction and its normal direction to the yield surface is shown in Figure 5 [12]. The normal direction angle $\theta_{2}$ to yield surface was approximately obtained by geometrical calculation. Taking a yield point B located on the yield locus as an example, first, the yield point B is connected with its adjacent yield points of A and C on the same yield surface; then, the intersection D is obtained by making the respective midperpendicular of the straight lines AB and BC, and finally, the vector $\vec{n}$ that passes through the intersection D and the yield point B is referred to as the normal direction at the yield point B, and the vector $\vec{n}$ representing plastic flow direction is orthotropic to the yield locus. The plastic flow direction angle $\theta_{1}$ to the yield surface is calculated by the following Equations (1)–(6). The vector of plastic strain increment in the $\sigma -\sqrt{3}\tau$ stress space can be expressed as follows:(1)$$\Delta {\vec{\varepsilon }}^{p}=\Delta \varepsilon_{1}^{p}{\vec{n}}_{1}+\Delta \varepsilon_{3}^{p}{\vec{n}}_{3}$$
(2)$$\Delta \varepsilon_{1}^{p}=\Delta \varepsilon^{p}=\Delta \varepsilon -\frac{\Delta \sigma }{E}$$
(3)$$\Delta \varepsilon =\varepsilon -\varepsilon_{0};\Delta \sigma =\sigma -\sigma_{0}$$
(4)$$\Delta \varepsilon_{3}^{p}=\frac{\Delta \gamma^{p}}{\sqrt{3}}=\frac{\Delta \gamma -\frac{\Delta \tau }{G}}{\sqrt{3}}$$
(5)$$\Delta \gamma =\gamma -\gamma_{0};\;\Delta \tau =\tau -\tau_{0}$$
where Δσ and Δτ are axial stress increment and shear stress increment, respectively; $\Delta \varepsilon^{p}$ and $\Delta \gamma^{p}$ are axial plastic strain increment and shear plastic strain increment, respectively; $\varepsilon_{0}$ and $\gamma_{0}$ are axial residual strain and shear residual strain at the unloading point, respectively; $\sigma_{0}$ and $\tau_{0}$ correspond to the axial stress and the shear stress at the unloading point, respectively; and E and G are elastic modulus and shear modulus, respectively.

The plastic flow direction in the $\sigma -\sqrt{3}\tau$ stress space is defined as
(6)$$\theta =\arctan(\frac{\Delta \varepsilon_{2}^{p}}{\Delta \varepsilon_{1}^{p}})$$

## 3. Constitutive Relationships and Coupling Methodology

### 3.1. Flow Rule and Strain Hardening Modeling

According to the classical multiplicative decomposition of the deformation gradient, the total deformation gradient F within the crystal plasticity framework can be decomposed into elastic and plastic components [40,41]:(7)$$F=F^{*}\cdot F^{p}$$
where the elastic component $F^{*}=R^{L}\cdot U^{e}$ is caused by the elastic stretching $U^{e}$ and rigid body rotation $R^{L}$ of a crystal lattice, and the plastic component $F^{p}$ describes the dislocation glide along the specific slip planes.

The velocity gradient L can be decomposed into the elastic gradient tensor $L^{*}$ and the plastic gradient tensor $L^{p}$ [40,41,42]:(8)$$L=\dot{F}\cdot F^{-1}=L^{*}+L^{p}={\dot{F}}^{*}\cdot F^{{*}^{-1}}+F^{*}\cdot {\dot{F}}^{p}\cdot F^{p^{-1}}\cdot F^{{*}^{-1}}$$
where $L^{*}={\dot{F}}^{*}\cdot F^{*}{}^{{}^{-1}}$ represents elastic deformation gradient and $L^{p}={\dot{F}}^{p}\cdot F^{p^{-1}}$ stands for plastic deformation gradient.

Further, because only the activated slip systems contribute to the plastic deformation in each grain, the plastic velocity gradient $L^{p}$ is expressed as the sum of shear strain rates ${\dot{\gamma }}^{\left(\alpha \right)}$ on all active systems [43]:(9)$$L^{p}={\dot{F}}^{p}\cdot F^{p^{-1}}=\sum_{\alpha =1}^{n}(m^{\left(\alpha \right)}⊗n^{\left(\alpha \right)}){\dot{\gamma }}^{\left(\alpha \right)}=\sum_{\alpha =1}^{n}s^{\left(\alpha \right)}{\dot{\gamma }}^{\left(\alpha \right)}$$
where ${\dot{\gamma }}^{\left(\alpha \right)}$ is the plastic shear rate along the crystallographic slip directions, and n is the total number of slip systems considering the typical 12 slip systems {111} < 110 > for a single crystal of the face centered cubic (FCC) structure materials. The unit vectors $m^{\left(\alpha \right)}$ and $n^{\left(\alpha \right)}$ stand for the slip direction and the normal direction with respect to the slip plane in the intermediate configuration, respectively; their dyadic product ($s^{\left(\alpha \right)}=m^{\left(\alpha \right)}⊗n^{\left(\alpha \right)}$) defines the Schmid tensor.

The crystalline lattice rotation will lead to the transformation of these vectors $m^{\left(\alpha \right)}$ and $n^{\left(\alpha \right)}$ in the reference configuration; further, the corresponding vectors $m^{(\alpha )*}$ and $n^{(\alpha )*}$ in the current configuration can be obtained by the following equations:(10)$$m^{(\alpha )*}=F^{*}\cdot m^{\left(\alpha \right)}$$
(11)$$n^{(\alpha )*}=n^{\left(\alpha \right)}\cdot F^{{*}^{-1}}$$
where the vectors $m^{\left(\alpha \right)}$ and $n^{\left(\alpha \right)}$ are assumed to be orthogonal to each other.
(12)$$m^{(\alpha )*}\cdot n^{(\alpha )*}=m^{\left(\alpha \right)}\cdot n^{\left(\alpha \right)}=0$$

### 3.2. Hardening Rules

The mesoscopic deformation mechanism of polycrystalline materials follows the deformation law of each individual crystal. Because the subsequent yield surfaces are studied under pre-cyclicloading, in order to address the influence of Bauschinger effect, a back-stress $x^{\left(\alpha \right)}$ is here introduced by Zhang [7] into the viscoplastic constitutive model. On the basis of the viscous regularization proposed by Hutchinson [44], the slip rate for a given slip system is written as
(13)$${\dot{\gamma }}_{\left(\alpha \right)}={\dot{\gamma }}_{0}\mathrm{sgn}\left(\tau^{\left(\alpha \right)}-x^{\left(\alpha \right)}\right){\left|\frac{\tau^{\left(\alpha \right)}-x^{\left(\alpha \right)}}{g^{\left(\alpha \right)}}\right|}^{k}$$
where ${\dot{\gamma }}_{0}$ is a reference shear strain rate on the α-th slip system, k is the rate sensitivity exponent, $\tau^{\left(\alpha \right)}$ is the resolved shear stress on the α-th slip system, $g^{\left(\alpha \right)}$ is the critical shear stress on the activated slip system α to govern the isotropic hardening of the crystal, and the back-stress $x^{\left(\alpha \right)}$ is especially introduced here to characterize the nonlinear directional hardening of the crystal on the α-th slip system.

The critical shear stress $g^{\left(\alpha \right)}$ evolves with plastic strain on a given activated slip system according to the following form proposed by Hill [45]:(14)$${\dot{g}}^{\left(\alpha \right)}(\gamma )=\sum_{\beta =1}^{n}h_{\alpha \beta }(\gamma )\left|{\dot{\gamma }}^{\left(\beta \right)}\right|;\;\gamma =\int \sum_{\beta }^{n}\left|d\gamma^{\left(\beta \right)}\right|$$
where ${\dot{\gamma }}^{\left(\beta \right)}$ is the slip rate on the system β, n is the total number of crystal slip systems, and γ is the accumulated shear strains on all active slip systems. $h_{\alpha \beta }(\gamma )$ is the slip hardening modulus matrix governing the interaction between various slip systems, which is suggested by Hutchinson [44] as follows:(15)$$h_{\alpha \beta }(\gamma )=h(\gamma )[q+(1-q)\delta_{\alpha \beta }]$$
where $\delta_{\alpha \beta }$ is the Kronecker delta, and q refers to the latent-hardening ratio that has values in the range of 0 to 1. A function describing the self-hardening ratio h(γ) is defined by Chang and Asaro [46] as follows:(16)$$h(\gamma )=h_{0}\sec h^{2}\left(\frac{h_{0}\gamma }{\tau_{s}-\tau_{0}}\right)$$

Where $h_{0}$ is the initial hardening rate, $\tau_{0}$ is the critical resolved shear stress, and $\tau_{s}$ is the saturation shear stress.

### 3.3. Back Stress Evolution

In order to model cyclic plastic behavior reasonably, on the basis of the models of Walker [47] and Chaboche [48], the evolution of back-stress ${\dot{x}}^{\left(\alpha \right)}$ is given by [49].
(17)$${\dot{x}}^{\left(\alpha \right)}=a{\dot{\gamma }}^{\left(\alpha \right)}-c\;[1-e_{1}(1-\exp(-e_{2}\gamma ))]x^{\left(\alpha \right)}\left|{\dot{\gamma }}^{\left(\alpha \right)}\right|-dx^{\left(\alpha \right)}$$
where a is the linear hardening constant of slip systems, c and d denote the nonlinear hardening constants, and $e_{1}$ and $e_{2}$ are constant parameters representing the law of the cyclic hardening or softening. The first term of the equation denotes a linear strain hardening term. The second one denotes a dynamic recovery term. The third one denotes a static recovery term. Moreover, the back-stress term can adequately cover the cyclic hardening, the Bauschinger, and similar strain path change effects.

The detailed constitutive equations of the crystal plasticity model have been calculated and implemented into the user-supplied subroutine UMAT from the ABAQUS finite element code [49].

## 4. Modeling

### 4.1. Finite Element Models

The global finite element model (GFEM) simulating the same thin-walled tube specimen as the experiments of yield surfaces was built, consisting of the 64,000 elements and 3600 grains in the center gauge region of the GFEM, and the grain distribution in the cross-section of gauge length is also shown in Figure 6, with the grain number of the GFEM being sufficient to reflect the macroscopic convergent stress–strain response shown in the previous study [50]. Each grain is assigned to random shape, size, and random crystal orientation [50]. The center gauge section of the specimen defines two reference points A and B, simulating a 25 mm axial-torsional extensometer to measure the macroscopic axial strain and torsion strain under the combine tension–torsion loading. The loading and boundary conditions of the GFEM model are as follows: all element nodes at the loading end or the fixing end (the region marked in the red dotted line in Figure 6 are coupled with the corresponding reference point to form a point set, respectively, and loads and boundary conditions are applied to the reference points). All six degrees of freedom for the reference point in the fixed clamping end were set as zero. An axial load **F** and a torsion load **T** (combined tension–torsion loading) were applied on the reference point in the load clamping end along the first axial and the fifth rotational degrees of freedom as the testing, and the remaining four degrees of freedom for the reference point were set to zero U_2_ = U_3_ = UR_2_ = UR_3_ = 0, as shown in Figure 6.

Computational information was introduced in detail as follows: two server-cluster computers were used with 32 CPU cores in the parallel computing environment. The total calculation time of 30 pre-cycles was 6.3 h for each cycle including 200 increments. The loading process simulation corresponding to different re-loading path with the “restart write” function can be started simultaneously under different unloading cases after 30 pre-cycles. Each re-loading path was calculated with 10 CPU cores, and the calculation time was about 45 min.

### 4.2. Parameter Calibration of Crystalline Plasticity Models

The crystal lattice of a pure aluminum is a face-centered cubic. The crystal plastic model describing the elastic and plastic behavior of a single crystal contained 14 parameters (3 elastic constants and 11 viscoplastic parameters), and each parameter had physical meaning. The detailed determination process is as follows: three anisotropic elastic constants $C_{11}$, $C_{12}$, and $C_{44}$ of single crystal aluminum are obtained by fitting elastic region of testing uniaxial tensile and torsion stress-strain curves [50]. $\tau_{0}$ and $\tau_{s}$ are the critical resolved shear stress and the saturation value, respectively, which depend on the elastic range between the points of unloading and re-yielding on the stable stress–strain hysteresis loop. a and c describe the strain hardening of the elastoplastic region on the hysteresis loop, and e_1_ and e_2_ are used to describe cyclic hardening behavior during the initial 10 cycles. The reference strain rate was ${\dot{\gamma }}_{0}=0.001/s^{-1}$. The rate sensitivity exponent was defined as k = 200 to describe the insensitive strain rate of the pure aluminum. By taking into account the fact that creep is ignored under cyclic loading, the nonlinear hardening constants was set to $d=0/s^{-1}$. The remaining material parameters *τ*_0_, *τ*_s_, *h*_0_, *a*, *c*, *e*_1_, and *e*_2_ of the above crystal viscoplastic model are preliminarily estimated from the tested cyclic hysteresis curves, and more accurate material parameters can be fitted by trial-and-error method with test curve as the target curve. The summary of the final calibrated material parameters is listed in Table 2. Figure 7 shows the comparison of experimental curves (black lines) and simulated curves (red lines) of 30 pre-cycles, which confirms that the simulation results using the GFEM can match with the tested curves well. This good agreement suggests that the constitutive model can reasonably describe the cyclic plastic behavior of metal materials.

### 4.3. Effects of Loading Direction on Anisotropic Hardening Characteristics

Following the same experimental procedures for the case of unloading to the point O_D_ after pre-cyclic tension-compression loading, we conducted these predictions along the reloading directions at angles from 0° to 180° in steps of 30° in different proportions of tension (or compression) and torsion in the (σ,$\sqrt{3}\tau$) plane by using the GFEM on the basis of the crystal plasticity theory introducing a back stress. Then, the simulated σ-ε curves were also compared with experimental observations, as shown in Figure 8. The results show that the anisotropic strain-hardening properties can be observed, and the trend towards strain hardening of the simulated curves was consistent with that of the experimental curves under all proportional tensile–torsion loading paths, which had slight difference despite the relative large error caused by the dispersion of the test results by multi-specimen method and the model error at 60° reloading paths. The above results indicate that the anisotropic hardening behavior of polycrystalline aluminum after deformation can be reasonably described by the crystal plasticity model introducing nonlinear strain hardening equation. Moreover, it can be clearly seen from Figure 8a,b that the plastic pre-deformation governed the strain hardening rate of the material, namely, an increase in the flow stress with increasing angle deviated from the pre-loading direction was correctly captured by the model as the experimental observations. In other words, the strain hardening rate was weak in the pre-loading direction compared with the other direction, as well as the higher hardening rate upon the reverse pre-loading direction. The directional hardening and distorted shape of the subsequent yield surfaces could be attributed to the developed anisotropic hardening of the materials. It should be pointed out that the difference between initial yield surface and subsequent yield surface was noticeable. After plastic pre-deformation, for example, axial tensile deformation, the conspicuous anisotropic hardening of material can be observed. Therefore, the evolution of the subsequent yield surfaces is focused on and discussed in the following section.

## 5. Results and Discussion

### 5.1. Initial Yield Surfaces and Plastic Flow Directions

The initial yield surfaces and its plastic flow directions were simulated under proportional tensile and torsional stress paths using the yield definitions with the offset strain of 20 μ, 200 μ, and 1000 μ by means of GFEM integrated into crystal plasticity theory, as illustrated in Figure 9. The initial yield surfaces of the ploycrystal aluminum exhibited approximately isotropic evolution and obeyed the Tresca yield criterion in the. *σ* − $\sqrt{3}\tau$. stress plane with arbitrarily specified offset strain in Figure 9a. The plastic flow directions of the initial yield surfaces were represented by the vector arrow denoting the plastic strain increment of 50 μ in the present study. The influence of different offset strain definitions on the plastic flow directions is evaluated in Figure 9b. When the offset strain was very small, the deviation angles between the plastic flow direction and normality direction regarding the initial yield surfaces had subtle differences, and with a larger offset strain definition, the directions of plastic strain increment were normal to the initial smooth yield surface, being not evidently affected by the offset strain definition, indicating that the plastic flow directions of the initial yield locus approximately followed the normality rule with any offset strain definition, as shown in Figure 9.

### 5.2. Subsequent Yield Surfaces under Five Unloading Cases

As illustrated in Figure 3 and Figure 4, the pre-loading, unloading, and reloading paths and the measured method of yield points for probing the subsequent yield surface are described in detail in Section 2.2. The radial light gray lines representing the different reloading paths are drawn in Figure 10. The subsequent yield surfaces were predicted by the crystal plasticity model associated with the GFEM containing 64,000 elements and 3600 crystalline grains under the five unloading paths with the offset strain of 20 μ, 50 μ, 100 μ, 200 μ, 600 μ, and 1000 μ, as shown in Figure 10. With reference to the hardening curves under different reloading paths in Figure 8, the curves appeared with the lower strain hardening rate in the pre-loading direction compared with the other direction and with higher hardening rate upon the reverse pre-loading direction. Such directional hardening behaviors related to loading paths resulted in the anisotropic expansion and distortion of subsequent yield surfaces. The yield surfaces with different offset strain definitions nearly overlapped in the pre-loading direction but they expanded greatly towards the reverse direction with the increase of the offset strains during the plastic deformation process. As seen in Figure 10, for the small offset strains between 20 and 200 μ, the subsequent yield surfaces were gradually altered from the extremely sharper corner to a rather blunt vertex in the pre-strained direction and remained flat in its reverse direction, whereas the subsequent yield surfaces evolved into a similar ellipse based on isotropic expansion with the larger offset strain of 600 and 1000 μ. As shown in Figure 10, following the variation of the pre-strain direction from tension to compression, the sharp vertex direction of the subsequent yield surfaces simultaneously changed to be consistent with it, and the subsequent yield surfaces were translated observably with it in stress space and were enlarged in contrast to the initial yield surfaces resulting from the previous cyclic hardening. Therefore, the size and shape of the subsequent yield surface depended on the offset strain level, direction of the accumulated plastic strain, and unloading point cases.

### 5.3. The Plastic Flow Direction of Subsequent Yield Surfaces

The plastic flow evolution of the subsequent yield surfaces was analyzed by the crystal plasticity simulation with the offset strain of 20 μ, 200 μ, and 1000 μ under five unloading cases, as shown in Figure 11. In addition, the comparison of between theoretical calculations and experimental results is shown in Figure 12. The vector arrows denoting the plastic flow direction were prescribed with the plastic strain increment of 50 μ. As shown in Figure 11 and Figure 12, the plastic flow direction of the subsequent yield surfaces directly depended on the applied offset strain levels and the pre-strain paths. By yielding definition of the smaller offset strain, the plastic strain increment vectors (red arrows) corresponding to each yield point calculated by the Equations (1)–(6) according to the simulated and experimental data showed larger deviation from a vector normal to the yield surfaces (black arrow) and simultaneously rotated away from the pre-loading direction, whereas when a large offset strain definition was used to determine the yield surface, negligible deviation was observed between the computed plastic flow direction and orthogonal direction, namely, the vectors of plastic strain increment were nearly orthogonal to the yield surface, indicating that normality flow rule of a smooth yield surface was obeyed. The largest deviation of the plastic flow direction from orthogonal direction appeared in the inverse pre-strain direction, but the difference between them in the pre-strain direction was not significant.

### 5.4. Statistical Analysis for Plastic Flow Direction of Subsequent Yield Surfaces

By statistical analysis, the deviation angles $\theta_{2}-\theta_{1}$ between the plastic flow direction and the normal direction of the yield surfaces calculated in terms of the simulated and experimental data under the two unloading paths with the offset strain of 20 μ, 200 μ, and 1000 μ are shown in Figure 13. The results show that the plastic flow directions of subsequent yield surfaces directly depended on the applied offset strain levels. The deviation angles $\theta_{2}-\theta_{1}$ of subsequent yield surfaces became small as the offset strain proceeded. The plastic flow direction was associated with the shape of the subsequent yield surfaces; furthermore, the more severe anisotropy of the yield surfaces with a sharp vertex resulted in the larger deviation from the normal direction with a small offset strain, whereas the normal flow rule was approximatively satisfied for the smooth yield surface with a larger offset strain, for which deviation angle $\theta_{2}-\theta_{1}$ was less than 5°. In the case of the same path and yield definition, the slight difference of deviation angle $\theta_{2}-\theta_{1}$ between the simulated results and experimental ones was mainly derived from the experimental yield surfaces determined by the fewer yield points; nevertheless, the trends of plastic flow direction by the crystal plasticity simulation were consistent with the experimental results.

## 6. Conclusions

In order to characterize the plastic anisotropy of metal deformation, we employed the crystal plasticity model introducing a back-stress to simulate the evolution of the subsequent yield surface of polycrystalline aluminum after pre-cyclic loading, with the simulated process being the same as the experimental approaches. At the same time, the plastic flow that reflects the direction of the plastic strain rate and a strain-hardening rule describing the yield surface evolution during plastic deformation of materials were also further studied, and the experimental observations and the simulation results were compared and analyzed. The main conclusions in this study are as follows:The crystal plasticity theory introducing a back-stress in conjunction with the GFEM provided the accurate simulated results with respect to the remarkable sharp corner and the non-associated flow direction of the subsequent yield surfaces as experimental observations under all explored cyclic pre-loading paths.For the arbitrarily specified offset strain definition, the initial yield surfaces of the ploycrystal aluminum exhibited approximate isotropic evolution and obeyed the Tresca yield criterion, noting that the performance was different from the Von Mises criterion predicted by the common software, in the $\sigma -\sqrt{3}\tau$ stress plane, and the plastic flow directions approximatively obeyed the normal flow rule.The evolution and plastic flow directions of the subsequent yield surfaces strongly depended on the offset strain levels. The subsequent yield surfaces defined with the smaller offset strain exhibited more severe distortion and larger deviation angles from the normal direction in comparison with those results using the larger offset strain. However, with the larger offset strain, the subsequent yield surfaces were similar to the convex ellipse, and the plastic flow directions approximatively obeyed the normal flow rule.The dependency of plastic flow on the anisotropic evolution of subsequent yield surfaces found that the normality rule of plasticity followed the smooth yield locus; conversely, the significant non-associated flow was attributed to the distorted yield locus. Furthermore, it is necessary to study the subsequent yield surfaces in depth and the plastic flow behaviors under multi-axial proportional and non-proportional cyclic loading conditions with different cycles in our future work.

## Figures and Tables

**Figure 1 nanomaterials-11-02397-f001:**
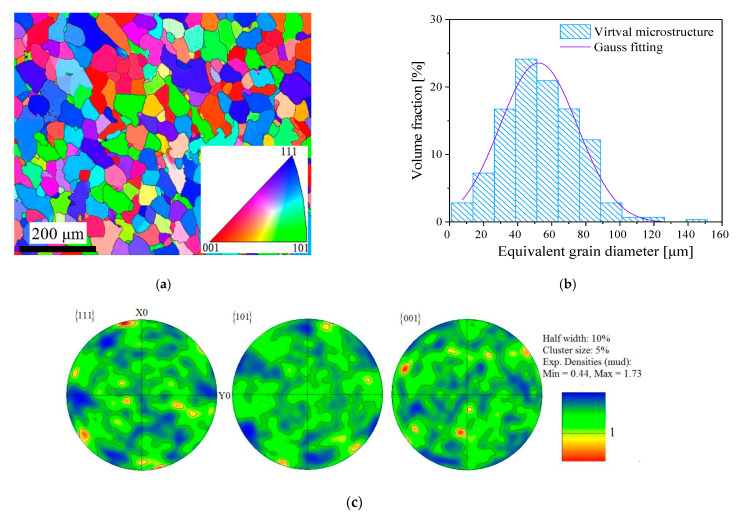
Cross-sectional microstructure of the specimen for pure aluminium observed by EBSD: (**a**) grain orientation map of pure aluminium; (**b**) the statistical distribution of the grain size; (**c**) the pole figure.

**Figure 2 nanomaterials-11-02397-f002:**
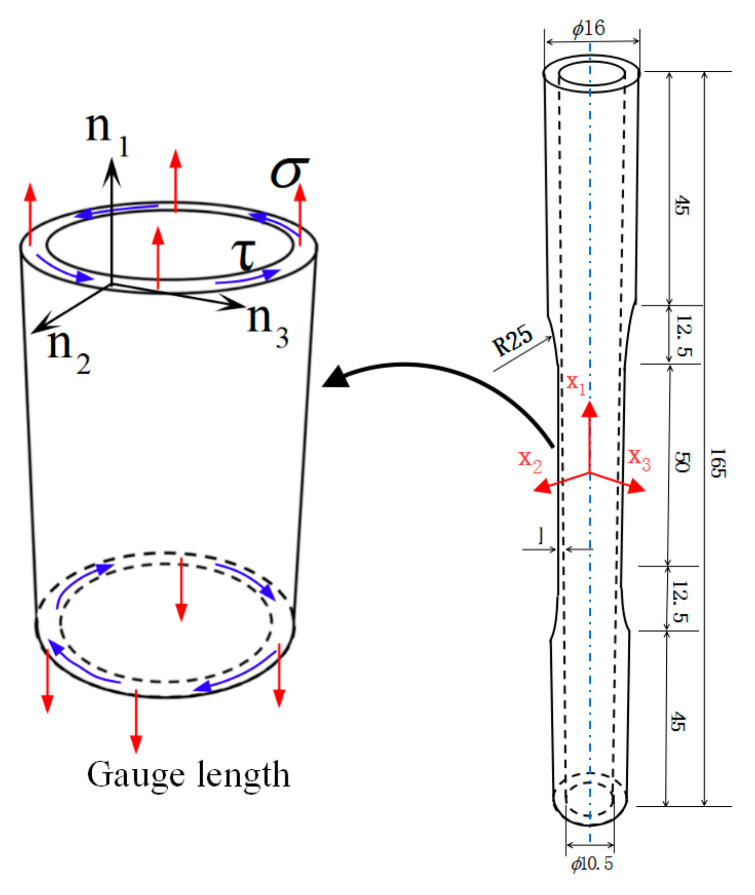
The geometrical figure for thin-walled tube specimens of pure aluminum (unit: mm).

**Figure 3 nanomaterials-11-02397-f003:**
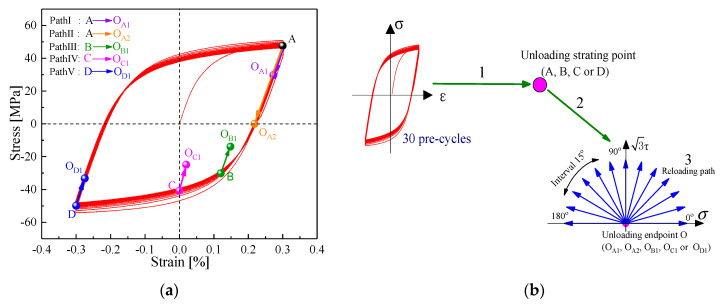
The probing paths of the subsequent yield surface including pre-cycle, unloading, and reloading cases: (**a**) different unloading starting points and end positions after tension-compression pre-cycle; (**b**) schematic diagram of pre-cycle, unloading, and reloading paths of yield surface.

**Figure 4 nanomaterials-11-02397-f004:**
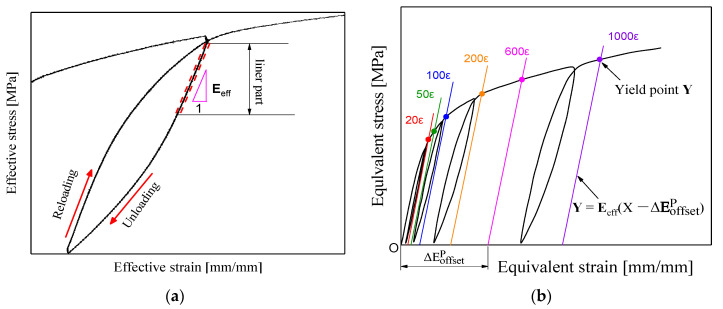
The yield point definition in terms of unloading modulus: (**a**) the determination method of unloading stiffness; (**b**) the variation of unloading stiffness **E**_eff_ with the increasing offset strain during reloading and unloading processes.

**Figure 5 nanomaterials-11-02397-f005:**
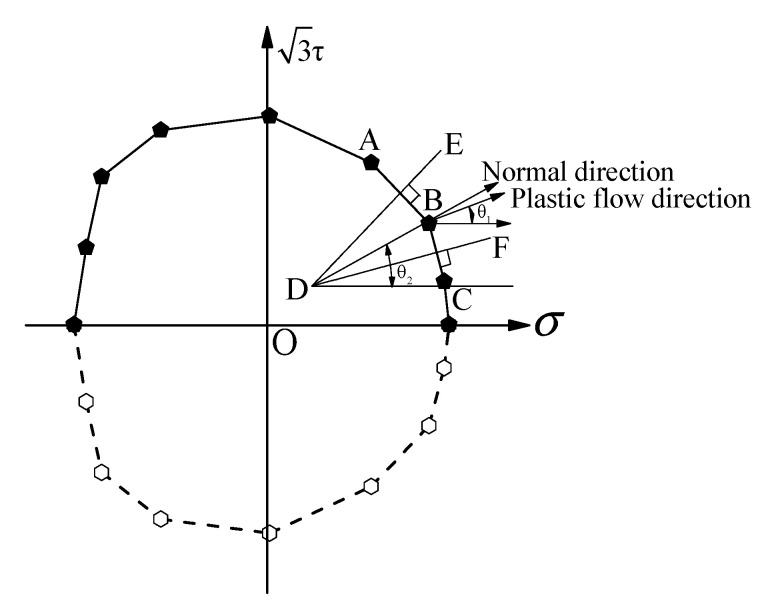
The schematic illustration about the determination of normal direction of yield point on yield surface and plastic strain incremental direction.

**Figure 6 nanomaterials-11-02397-f006:**
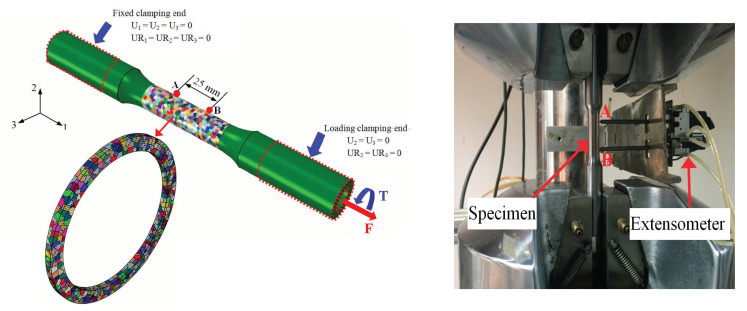
The loading process of the global finite element model (GFEM) containing 3600 grains under the combined tension–torsion loading.

**Figure 7 nanomaterials-11-02397-f007:**
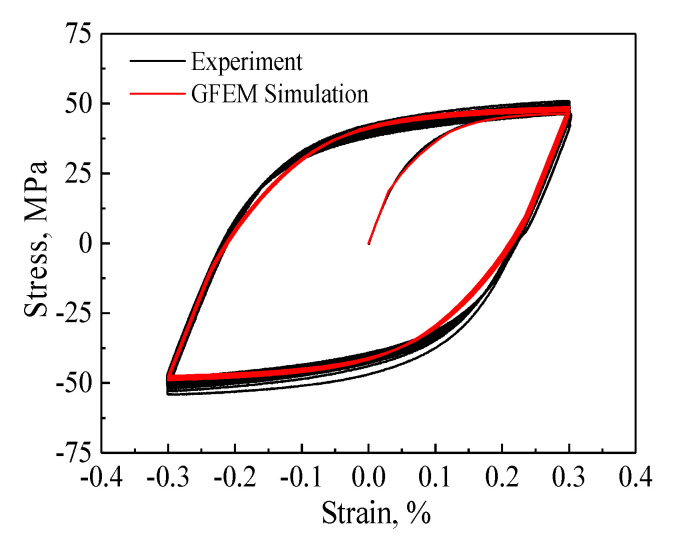
The comparison of the cyclic stress–strain curves under tension-compression loading by test and crystal plasticity simulation with the GFEM containing 3600 grains.

**Figure 8 nanomaterials-11-02397-f008:**
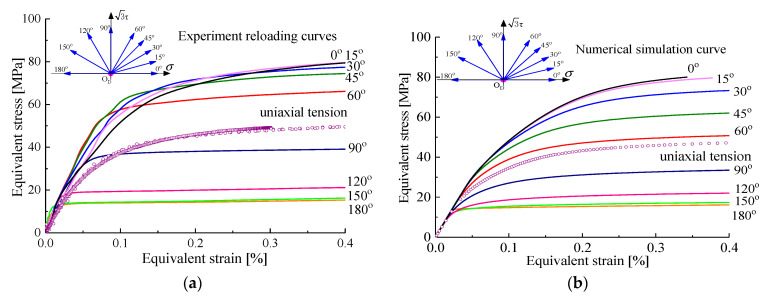
The strain-hardening curves under different tension–torsion reloading paths unloaded from the hysteresis loop D (−0.3%) to O_D_ after 30 pre-cycles by test and crystal plasticity simulation: (**a**) the experimental strain-hardening curves; (**b**) the simulated strain-hardening curves.

**Figure 9 nanomaterials-11-02397-f009:**
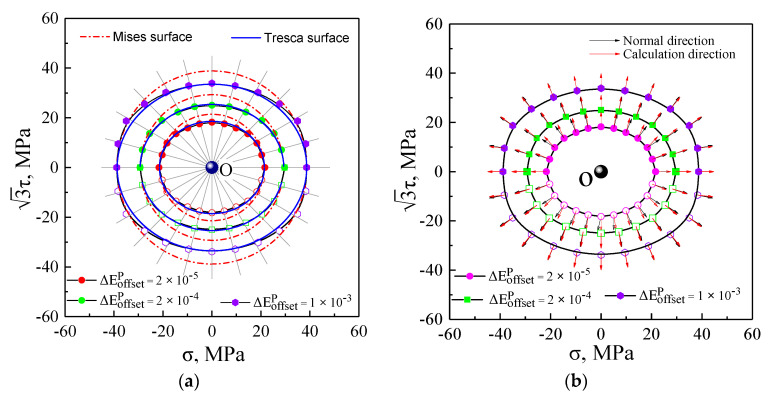
Initial yield surface by crystal plasticity simulation: (**a**) initial yield surfaces; (**b**) plastic flow directions.

**Figure 10 nanomaterials-11-02397-f010:**
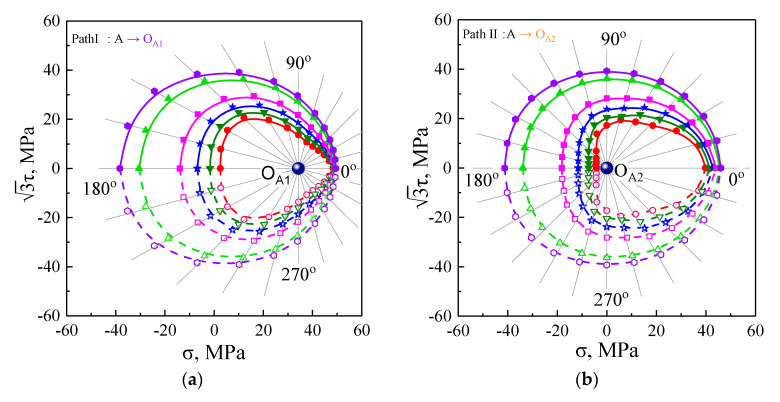
Subsequent yield surface at five different unloading paths by crystal plasticity simulation: (**a**) from A to O_A1_; (**b**) from A to O_A2_; (**c**) from B to O_B_; (**d**) from C to O_C_; (**e**) from D to O_D_.

**Figure 11 nanomaterials-11-02397-f011:**
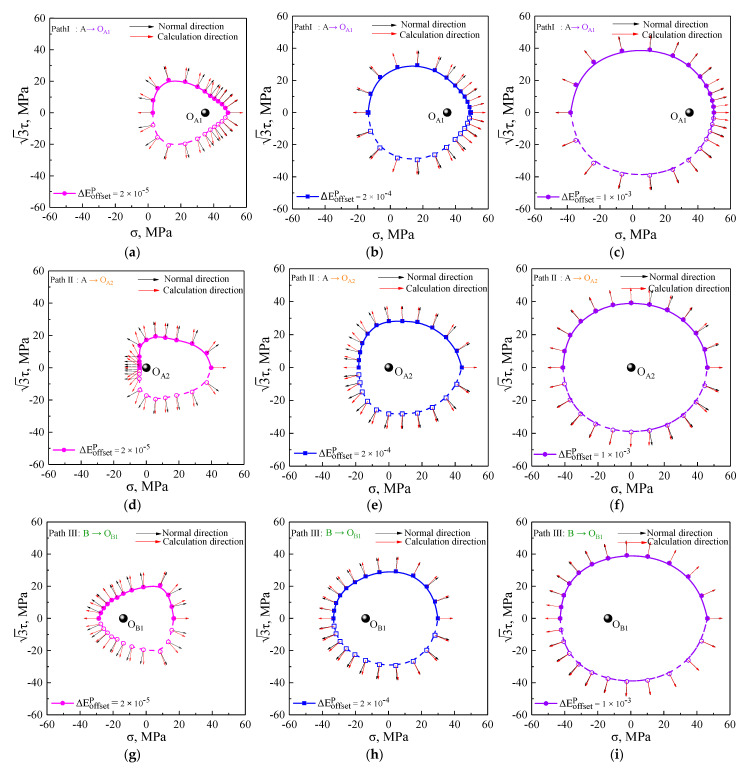
The plastic flow directions by theoretical calculations and simulated results with the offset strain of 20 μ, 200 μ, and 1000 μ under five unloading cases: (**a**–**c**) at unloading points O_A1_; (**d**–**f**) at unloading points O_A2_; (**g**–**i**) at unloading points O_B1_; (**j**–**l**) at unloading points O_C1_; (**m**–**o**) at unloading points O_D1_.

**Figure 12 nanomaterials-11-02397-f012:**
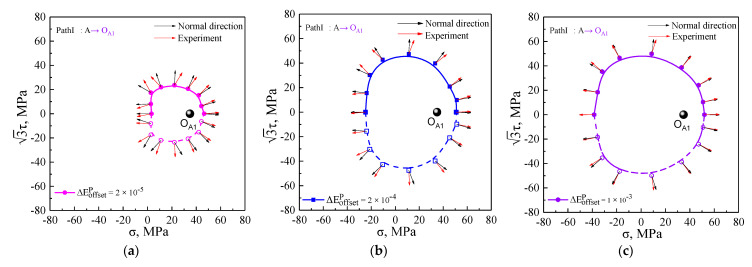
Comparison of plastic flow directions between theoretical calculations and experimental test at offset strain of 20 μ, 200 μ, and 1000 μ under five unloading cases: (**a**–**c**) at unloading points O_A1_; (**d**–**f**) at unloading points O_A2_; (**g**–**i**) at unloading points O_B1_; (**j**–**l**) at unloading points O_C1_; (**m**–**o**) at unloading points O_D1_.

**Figure 13 nanomaterials-11-02397-f013:**
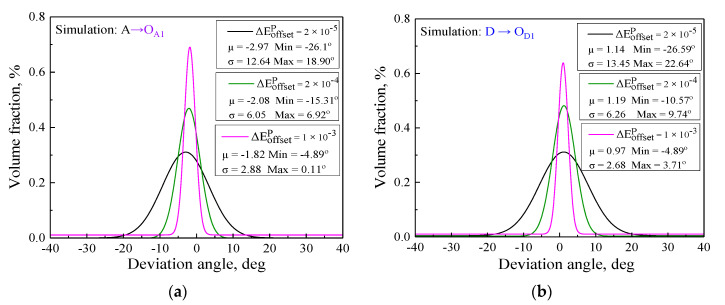
The statistical distribution of the deviation angles between the plastic flow direction and the normal direction of the yield surface: (**a**) simulated statistical distribution at unloading points O_A1_; (**b**) simulated statistical distribution at unloading points O_D1_; (**c**) experimental statistical distribution at unloading points O_A1_; (**d**) experimental statistical distribution at unloading points O_D1_.

**Table 1 nanomaterials-11-02397-t001:** Chemical compositions and mechanical properties of polycrystalline aluminum.

Chemical Composition						Mechanical Properties				
Al/%	Cu/%	Mg/%	Si/%	Mn/%	Zn/%	E/GPa	G/GPa	$\sigma_{0.2}/\mathrm{MPa}$	$\sigma_{u}/\mathrm{MPa}$	$\varepsilon_{f}$
99.89	0.02	0.03	0.03	0.02	0.01	57	25	20	81	24%

**Table 2 nanomaterials-11-02397-t002:** The material parameters of single crystal aluminum.

$C_{11}$	$C_{12}$	$C_{44}$	$\tau_{0}$	$\tau_{s}$	$h_{0}$	a	c	$e_{1}$	$e_{2}$	d	${\dot{\gamma }}_{0}$	k
/GPa	/GPa	/GPa	/MPa	/MPa	/MPa	/GPa	/GPa			/s^−1^	/s^−1^	
75.64	36.86	25.4	8.42	9.2	65	22.5	2.5	0.1	1.0	0	0.001	200

## Data Availability

The data presented in this research are available on request from the corresponding author.

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
