# Peer review of "Investigation on Plastic Flow Behaviors of FCC Polycrystalline Aluminum under Pre-Cyclic Tension-Compression Loading: Experiments and Crystal Plasticity Modeling"

_nanomaterials, 2021, doi:10.3390/nano11092397_

Round 1

Reviewer 1 Report

The authors use a CPFEM framework to predict the anisotropic plastic response of coarse-grained aluminium (and therefore, strictly speaking, not a nanomaterial), also after a broad gamut of strain path changes. The results have been extensively validated with appropriate experiments. This is a very valuable contribution in itself, in particular with regard to fatigue applications.

However, the power of the CPFEM approach is not really exploited, since no intragranular microstructures or strain gradients are considered. Instead, a regular (matrix) strain hardening law is used on slip system level, and a phenomenological back (shear) stress term is added there to cover the Bauschinger and similar strain path change effects. First, this addition should be motivated and discussed, including its shortcomings. Second, like this, the contribution of the grain morphology as characterized with EBSD can not be predicted, and the contribution due to intragranular microstructure and strain gradients can only be covered to the extent that they are compatible with the pre-concept of dislocation pile-ups. Only the contribution due to crystallographic texture is fully covered, but for that purpose a (much faster) statistical model is perfectly sufficient and no CPFEM is needed. 

A few technical remarks:

1.) The initial texture should be presented in terms of an ODF or a pole figure or, if it is a weak texture, of a statement on its maximum intensity in how many times random. 

2.) The fitting procedure used to identify the material parameters should be described and the obtained values be compared to literature values from experiment and other simulations, if available, and be discussed. 

3.) For a proper assessment of the proposed model, it should best also be applied to "less ideal" microstructures and textures, e.g. to a fine grained and/or a strongly textured material. 

4.) In the literature survey, some key papers on CPFEM, also on applications to forming processes involving strain path changes, are missing, e.g. the review papers by Roters et al. 2010 in Acta Materialia or 2019 in Computational Materials Science (and no, this referee is not from that school).

5.) Computational information on the used machine(s) and the required CPU time should be added. 

6.) In the conclusions, the authors mention that "the crystallography of slip and texture ... will be addressed in [their] future works" - I thought to have understood from the manuscript that a crystal plasticity model is embedded as UMAT in ABAQUS and used for the present work, i.e. that the influence of crystallography and texture ARE already covered??

Author Response

Thank you for valuable comments on our article.We have made the modification in revisied manuscript according to your suggestion, please see the attachment.

Reviewer 2 Report

The classical theory of plastic flow is widely used in the calculations of technological processes, including in common packages (ANSYS, ABAQUS, etc.), which is due to the acceptable accuracy of the results and high computational efficiency of the models created on its basis. One of the difficulties in its application is the need to establish an evolving yield surface at each deformation moment, which requires performing experiments on complex loading (generally speaking, in five-dimensional combined stress-strain spaces). At present, full-scale experiments can be carried out in spaces of dimension no higher than 3 (tubular specimens, loading by axial forces, torque and internal pressure). With the advent of multilevel physically oriented models (such as the crystal plasticity finite element method (CP FEM)), attempts are being made to use these models as a “theoretical testing machine”. The proposed article also discusses the possibility of using CP FEM to reveal the laws governing the evolution of the yield surface for sheet materials subjected to preliminary plastic deformation. It should be noted that, in contrast to field experiments, "theoretical experiments" allow a more detailed understanding of the physics of the process, the main mechanisms of plastic deformation. The above allows us to consider the research topic chosen by the authors as being of certain interest; at the same time, it seems necessary to substantiate the relevance of the proposed work in more detail.

There are some questions and comments on the content of the article.

  1. Researchers - mechanics have been engaged in the formulation of constitutive relations (CR) for describing plastic deformation for more than 100 years. They have proposed a large number of approaches and particular theories. In this regard, it is desirable to mention in the review various classes of CR (for example, the endochronic theory of K. Valanis, the theory of elastoplastic processes by A.A. Ilyushin, etc.), to justify the choice of the classical theory of plastic flow as the basic one. It should be noted that deformable solids are materials with slowly decaying memory; therefore, the change in the yield surface depends significantly on the history of deformation, and, therefore, a specific law of evolution of the yield surface will correspond to each investigated deformation process. In other words, to describe the evolution of the yielding surface, many cardinality of the continuum of laws may be needed. In this connection, it seems appropriate to more clearly formulate the goal and tasks of the study, to determine the class of loading processes for which it is planned to construct modifications of the relations of the theory of plastic flow.
  2. Section 2.2. It is necessary to explain the choice of loading programs. Generally speaking, the theory of plasticity is the basic theory for determining the technological regimes of metal forming, in which materials undergo very large deformations (hundreds or more percent). It is not clear to what extent the regularities of the evolution of the yield surface obtained in the considered deformation ranges are suitable in the future for application to the solution of practically important problems.
  3. 2.3, papers 5-6. The article examines the properties of anisotropic material. The causes of anisotropy are usually structural changes in the material caused by the previous inelastic deformation, including the appearance of an uneven distribution of grain lattice orientations. In this regard, the question arises about the legality of using the isotropic law of elasticity (relations (2), (4)).

With the chosen method for determining the normal to the yield surface (Fig. 5), its direction significantly depends on the step of changing the angle (Fig. 3b). It is advisable to conduct additional research with a decrease in the step of changing the angle.

  1. 3.3, p.8. It is necessary to indicate the dimensions of the parameters, especially the constant d, which must have a dimension of s–1, in connection with which it is necessary to clarify its physical meaning.
  2. 4.1, p.9, fig.6. The boundary conditions shown in the figure are not provided with any explanations or designations, which makes them difficult to understand. If the computational domain is limited by a part of the sample, indicated by multi-colored finite elements, then it is necessary to give the boundary conditions for the boundaries of this particular area (three scalar conditions at each point of the boundary of this area). Judging by the above results, a rigid (kinematic) loading was specified; in this case, the question arises: how were the kinematic conditions set at the boundary of the computational domain, was the deformability of the part of the sample between the gripping region and the boundary of the computational domain taken into account?
  3. 4.3, p. 10. The text indicates that the difference between the calculated and experimental values ​​of the effective stresses does not exceed 10% (lines 285-286). At the same time, for a trajectory with an angle of 120 °, the differences significantly exceed the indicated interval, how can this discrepancy be explained?
  4. 5.1, p. 11, Fig. 9a). It is not clear how the Tresca yield surface is obtained; in the classical theory of plasticity, the Tresca criterion defines a ruled surface in the deviatoric stress space.
  5. What specific proposals can be formulated on the basis of the results obtained to improve the accuracy of the description of inelastic deformation processes using the constitutive relations of the classical theory of plastic flow?

There are some remarks on the terminology and presentation of the material

  • Page 1, line 38. It is more correct to speak not about elastic and plastic regions, but about elastic and elastoplastic regions, since even in the case of an ideal plastic material, both plastic and elastic deformations change during active deformation.
  • Pages 3-4, lines 122-123, Fig. 2: there is a discrepancy between the description of the basis vectors n1, n2, n3 in the text and their image in the figure.
  • S.2.2, line 143 and further. It is necessary to give all the designations used in the article, especially if they differ from the widely used ones. It is not clear what is meant by the notation μ, ε; in most works on the mechanics of a deformable solid, μ denotes the shear modulus, ε denotes the value of uniaxial deformation (and in this capacity it is used below). According to those shown in Fig. 9 and further results, it can be assumed that these values, together with a numerical factor, determine the plastic deformation tolerance used to establish the conditional yield stress, but the designations should hardly serve as an object of guesswork for the reader.
  • Page 6, lines 171-172. The reference to Fig. 2 is not clear, this figure shows only the designations and dimensions of the sample, no data on the distribution of stresses over the working part of the sample are presented.
  • S.5.1, p.10, line 309: what does the abbreviation GLBM mean?

Author Response

Thank you for  valuable comments on our article. We have made corrections in the manuscript according to your comments. please see the attachment.

Reviewer 3 Report

The present work analyzes the plastic anisotropy of deformation in polycrystalline aluminum both computationally and experimentally. A crystal plasticity involving a back-stress is here employed to simulate the evolution of the yield surface of the metal after a pre-cyclic loading, whose model is successfully verified in its reliabity against experimental predictions stemming from statistical methods. Based on results and conluding remarks, the work can have a great impact on the scientific community working in this field, and is worthy of publication. A minor revision is required before accepting  the work definitively for publication, due to the presence of some sparsed typos in the text, that must be properly checked and corrected.

Author Response

Thank you for your valuable comments on our article. We have checked and corrected these errors in the text. please see the attachment.

Round 2

Reviewer 2 Report

Point 1. The authors' answers to the remarks given in the first item are not very clear, perhaps the reviewer has not formulated them clearly enough. The possibility of constructing an evolving yield surface using CP FEM was established in many works of the 70s – 80s of the XX century. Research in this direction makes sense only in the case of the development of modified theories of plastic flow, which are more universal than the existing constitutive relations (CR). The main question was what exactly the authors set as their goal in improving the classical theory of plastic flow?

The works of K. Valanis and A.A. Ilyushin were initially focused on the construction of theories of plasticity that did not use the concept of a yield surface. The emergence of a yield surface in the theory of K. Valanis is a consequence of the theory and rather indicates a violation of the initially declared thesis about the absence of this surface.

Points 2, 9. In fact, these items are adjacent to the previous one. For a particular type of loading, the authors showed the features of the transformation of the yield surface. Is it possible to formulate proposals for the formulation of the law of evolution of the yield surface for other loads?

Point 6. The text lacks the designations Ui, URi, i = 1,2,3. In the formulation of boundary value problems in continuum mechanics, the boundary conditions are formulated for each material point (particle) of the boundary of the investigated body. Material points have only three degrees of freedom, it is not clear what 6 degrees of freedom are we talking about.

On points 3–5, 7–8, 10–14, the authors' answers can be considered satisfactory.

Author Response

Please see the attachment,last modification was highlighted in yellow ,the current modification is marked  in red.
